Discovery of novel astrovirus genotype species in small ruminants

Kauer Ronja V. 1
http://orcid.org/0000-0002-3814-5067 Koch Michel C. 1 2
Hierweger Melanie M. 1 2
Werder Simea 1
http://orcid.org/0000-0001-5503-2649 Boujon Céline L. 1 2
Seuberlich Torsten 1 torsten.seuberlich@vetsuisse.unibe.ch
1 Division of Neurological Sciences, Vetsuisse Faculty, University of Bern , Bern , Switzerland
2 Graduate School for Cellular and Biomedical Sciences, University of Bern , Bern , Switzerland
Spilki Fernando
Electronic publication date: 2019 Jul 31
Publication date: 2019
Volume: 7
Electronic Location ID: e7338
Received 2019 Feb 27; Accepted 2019 Jun 23
Copyright: © 2019 Kauer et al.
Copyright year: 2019
Copyright holder: Kauer et al.
License: This is an open access article distributed under the terms of the Creative Commons Attribution License, which permits unrestricted use, distribution, reproduction and adaptation in any medium and for any purpose provided that it is properly attributed. For attribution, the original author(s), title, publication source (PeerJ) and either DOI or URL of the article must be cited.
License URL: https://creativecommons.org/licenses/by/4.0/

Keywords: Astrovirus, Metagenomics, Virus discovery, Small ruminants, RT-PCR-screening, Interspecies transmission

Funding: Swiss National Science Foundation 31003A 163438 Swiss Veterinary and Food Safety Office This work was supported by the Swiss National Science Foundation [No. 31003A 163438] and the Swiss Veterinary and Food Safety Office. The funders had no role in study design, data collection and analysis, decision to publish, or preparation of the manuscript.

==============================
Astroviruses (AstV) are single-stranded, positive-sense RNA viruses, best known for causing diarrhea in humans and are also found in many other mammals; in those, the relevance in gastroenteritis remains unclear. Recently described neurotropic AstV showed associations with encephalitis in humans as well as in other mammals. In Switzerland, two different neurotropic AstV were identified in cattle, as well as one in a sheep. The high genetic similarity between the ovine and one of the bovine AstV strengthens the hypothesis of an interspecies transmission. In humans, AstV associated with encephalitis were found also in human stool samples, suggesting that in these patients the infection spreads from the gastrointestinal tract to the brain under certain conditions, such as immunosuppression. Whether a similar pathogenesis occurs in ruminants remains unknown. The aims of this study were (1) the investigation of the potential occurrence of neurotropic AstV in feces samples, (2) the discovery and analysis of so far unknown AstV in small ruminants and other ruminant species’ fecal samples and (3) the examination of a potential interspecies transmission of AstV. To achieve these aims, RNA extraction out of 164 fecal samples from different ruminant species was performed and all samples were screened for known neurotropic AstV occurring in Switzerland, as well as for various AstV using RT-PCR. Positive tested samples were submitted to next generation sequencing. The generated sequences were compared to nucleotide- and amino acid databases, virus properties were identified, and phylogenetic analyses as well as recombination analysis were performed. The excretion of neurotropic AstV in small ruminants’ feces could not be demonstrated, but this work suggests the first identification of AstV in goats as well as the discovery of multiple and highly diverse new genetic variants in small ruminants, which lead to a classification into novel genotype-species. Additionally, the prediction of multiple recombination events in four of five newly discovered full or almost full-length genome sequences suggests a plausible interspecies transmission. The findings point out the occurrence and fecal shedding of previously unknown AstV in sheep and goats and pave the way towards a better understanding of the diversity and transmission of AstV in small ruminants.

Introduction

Astroviruses (AstV) are non-enveloped, single-stranded positive-sense RNA viruses with an icosahedral virion structure, appearing as a star-like shape in electron microscopy (Caul & Appleton, 1982). The AstV genome is 6.2–7.8 kb in size and polyadenylated at the 3′ end. It presents at least three open reading frames (ORF): ORF1a, ORF1ab, and ORF2. ORF1a and ORF1ab encode nonstructural precursor proteins, nsp1a, and nsp1ab. The latter is translated via a ribosomal frameshift mechanism, where ORF1b is translated together with ORF1a (Marczinke et al., 1994). ORF2 encodes the capsid precursor protein, which is then intra- and extracellularly further processed to mature structural proteins (Willcocks et al., 1994).

According to the affected host class, two AstV genera were established: Mamastroviruses (MAstV) representing genotype species affecting mammalian species and Avastroviruses containing those viruses found in avian species. Due to the availability of high throughput next-generation-sequencing (NGS) technologies and the use of broadly reactive Pan-AstV RT-PCR protocols, there has been a remarkable increase in the number of AstV discovered in diverse species during the last years (Boujon, Koch & Seuberlich, 2017). AstV were first described in 1975 in a human stool sample (Appleton & Higgins, 1975; Madeley & Cosgrove, 1975). In humans, AstV are best known as a major source of outbreaks of gastroenteritis, especially in infants, young children and immunocompromised people (De Benedictis et al., 2011; Fischer, Pinho Dos Reis & Balkema-Buschmann, 2017). However, intestinal tissue infected with AstV shows only minor histological changes such as a mild intestinal inflammatory response (Sebire et al., 2004) and the knowledge on the pathogenesis of gastroenteric disease associated with AstV is still limited (Moser & Schultz-Cherry, 2005).

In 2010, AstV were found for the first time in association with encephalitis in a child with immunodeficiency (Quan et al., 2010). Thereafter, several novel AstV genotype species were detected in other human encephalitis cases (Brown et al., 2015; Lum et al., 2016). Encephalitis-associated AstV could be detected in stool samples, as well as in other body fluids, such as cerebrospinal fluid and plasma, suggesting that in these patients the infection spreads from the gastrointestinal tract to the brain (Cordey et al., 2016).

In animals, the state of knowledge about the tissue tropism of AstV is even more limited. Even though the presence of ovine astroviruses (OvAstV) in fecal sheep samples constituted the first report of AstV in animals (Snodgrass & Gray, 1977), still little is known about AstV infections in small ruminants, their transmission within and across species as well as their association with disease. In recent years a wide variety of mammalian domestic animal species were found positive for AstV in their feces; for example, cattle (Woode & Bridger, 1978), sheep (Snodgrass & Gray, 1977), red deer (Tzipori, Menzies & Gray, 1981), takins (Guan et al., 2018) and also domestic carnivores (Hoshino et al., 1981; Williams, 1980), mice (Kjeldsberg & Hem, 1985), and pigs (Bridger, 1980), but their role in the context of disease remained largely unclear. Remarkably, almost at the same time as the discovery of the first AstV-associated encephalitis in humans, the so-called shaking mink syndrome was described, which could be traced back to neurovirulent AstV infection (Blomstrom et al., 2010). One year later, in 2011, a neurovirulent porcine AstV type 3 could be identified as the cause of disease in outbreaks of meningoencephalomyelitis in piglets (Arruda et al., 2017; Boros et al., 2017).

Since 2013, different novel AstV genotype species were found as a plausible cause of non-suppurative encephalitis in cattle (Bouzalas et al., 2014; Li et al., 2013; Schlottau et al., 2016) and a few years later also in sheep (Pfaff et al., 2017). In Switzerland, three neurotropic AstV were identified in brain-tissue of ruminants; bovine astrovirus CH13 (BoAstV-CH13) and bovine astrovirus CH15 (BoAstV-CH15) in cattle (Bouzalas et al., 2014; Seuberlich et al., 2016), as well as ovine astrovirus CH16 (OvAstV-CH16) in sheep. The capsid protein as well as the non-structural proteins of this encephalitis-associated AstV in sheep (OvAstV-CH16) show a high similarity—around 99% on both the nucleotide and the amino acid level—to BoAstV-CH15, suggesting interspecies transmission of this genotype species between sheep and cattle (Boujon et al., 2017).

To date, there are 19 genotype species of Mamastrovirus (MAstV 1–19) recognized by the International Committee on Taxonomy of Viruses (ICTV). In particular, in OvAstV, little is known about their diversity. OvAstV-1 belongs to MAstV 13 and is the only enterotropic AstV closely related to neurotropic strains, but their exact taxonomy is still pending (Boujon, Koch & Seuberlich, 2017). Based on phylogenetic analyses of different viral strains of bovine, ovine and porcine origin, further evidence of possible interspecies transmission could be found (Donato & Vijaykrishna, 2017). The close clustering of farmed animals’ AstV strains reinforce the assumption of probable interspecies transmission events.

The aims of the present study included the assessment of a potential shedding of neurotropic AstV in fecal samples, the investigation of diverse AstV in different ruminant species and the examination of a potential interspecies transmission. Fecal samples of sheep, goats, deer, alpaca, and llamas were tested for BoAstV-CH13 as well as BoAstV-CH15/OvAstV-CH16 and screened for other AstV using a Pan-AstV RT-PCR. NGS and bioinformatics were used to recover viral genome sequences and to perform a phylogenetic comparison as well as a recombination analysis with other known AstV.

Materials and Methods

Samples

Fecal samples submitted for parasitological diagnostics were kindly donated by the Institute of Parasitology, Vetsuisse Faculty, University of Bern (Bern, Switzerland). In total 164 fecal samples, derived from 56 sheep, 56 goats, 30 alpacas, 12 deer, four llamas, two bisons, two chamois, one giraffe, and one ibex were investigated. Additional information on the animals and their health status was not available. Fecal samples were suspended 1:10 (w/v) in sterile PBS (137 mM NaCl, 10 mM Phosphate, 2.7 mM KCl; pH 7.4) and stored at −80 °C until further analysis. BoAstV-CH13 and OvAstV-CH16 positive and negative brain tissues were used as positive and negative controls for RT-PCRs and were available from the archive of the Division of Neurological Sciences, Vetsuisse Faculty, University of Bern (Bern, Switzerland). The summary of the laboratory workflow is shown in Fig. 1.

Figure 1 Virus discovery workflow in small ruminants’ fecal samples.

RNA extraction

All RNA extractions from fecal suspensions for AstV-screening by RT-PCR were done with the QIAamp Viral RNA Mini Kit (Qiagen, Hilden, Germany) and RNA isolation from brain tissue was performed with TRI Reagent (Sigma-Aldrich, St. Louis, MO, USA) according to the manufacturers’ protocols. To prepare samples for NGS, 500 μL of fecal suspensions were centrifuged at 16,000×g for 3 min. The supernatants were then centrifuged through Vivaclear MINI Clarifying filters with 0.8 μm PES (Sartorius, Göttingen, Germany) at 2,000×g for 5 min. Next, 280 μL of the filtrates were treated with 2 μL Benzonase (1U/μL) (Merck, Kenilworth, NJ, USA) for 2 h at 37 °C. The benzonase was inactivated by adding ethylenediaminetetraacetic acid (EDTA) to a final concentration of 5 mM. Finally, RNA extraction was performed with the QIAamp Viral RNA Mini Kit (Qiagen, Hilden, Germany) with the modification that the carrier RNA was omitted. RNA extracts were stored at –80 °C until further analysis.

Detection of neurotropic astroviruses BoAstV-CH13 and BoAstV-CH15/OvAstV-CH16 by RT-qPCR

To detect BoAstV-CH13, the RNA extracts were analyzed by either of two different probe based RT-qPCR assays (CH13-A or CH13-B) as described previously. While the CH13-A assay targets the 5′ part of the viral genome in ORF1a, the CH13-B assay detects the center of the viral genome in ORF2. Both protocols revealed a very similar excellent accuracy, precision, and analytical sensitivity (Lüthi et al., 2018).

For the detection of BoAstV-CH15 as well as OvAstV-CH16 a primer pair and a probe were designed based on full-genome sequence alignments of the following three strains from Switzerland and Germany; BoAstV-CH15, OvAstV-CH16 and BoAstV-BH89/14, targeting the 3′ end of ORF2 using the Geneious software package (Version 11.1.4; Biomatters, Auckland, New Zealand). The RT-qPCR was performed as described previously (Kuchler et al., 2019). After each cycle, the fluorescence was measured with the 6-carboxyfluorescein (FAM) channel and data was analyzed using the Sequence Detection Software (Version 1.4; Applied Biosystems, Foster City, CA, USA) with automatic baseline detection and a manual threshold of 0.2. The positive-negative cut-off was set at the cycle threshold (ct) value of 35. Each RT-qPCR run was performed using a positive and negative brain tissue control as well as a water control. All primers and probes are provided in Table S1.

Pan-astrovirus RT-PCR

For the detection of other AstV, a previously described heminested RT-PCR protocol using five degenerated primers; four forward and one reverse primer, which target a 450 nt long sequence at the 3′ end of ORF1b of a broad panel of AstV was used (Chu et al., 2008). Primer sequences are provided in Table S1. First-Strand cDNA synthesis was performed using the GoScript reverse transcriptase (Promega, Madison, WI, USA) and the gene specific reverse primer. The heminested PCR was done with a GoTaq Green Master Mix System (Promega, Madison, WI, USA) in two reaction rounds. For the first round, the PCR was set up in 25 μL reactions containing 12.5 μL of 2× GoTaq Green Master Mix, 4.5 μL of cDNA and a mixture of two forward primers PanAstV_forward 1 and −2 and the PanAstV_reverse primer, each in final concentration of 10 μM (Chu et al., 2008). The PCR was carried out with the following setting: 2 min, 95 °C and 30 cycles each of 30 s, 95 °C; 30 s, 50 °C; 30 s, 72 °C and final elongation 7 min, 72 °C. For the second round, the same reverse primer PanAstV-reverse was used as well as a mixture of PanAstV_forward_nested 1 and −2 (final concentration of 10 μM) and 1 μL of the first-round PCR-product as a template. Temperature settings were the same as in the first PCR, but with 40 cycles instead of 30 cycles.

Next generation sequencing

Prior to NGS, libraries were prepared using TruSeq DNA Nano Kit (Illumina, San Diego, CA, USA). For cDNA synthesis and polyA-selection the SMARTer Ultra Low Input RNA Kit (Takara Bio Inc., Kusatsu, Japan) was applied. NGS was performed on an Illumina HiSeq 3000 or NovaSeq 6000 in paired-end mode (2 × 150 bp).

Bioinformatics analysis

Reads were quality-trimmed with trimmomatic (Version 0.36) and mapped to their respective host genomes (alpaca: BioProject PRJNA30567, assembly Vicugna_pacos-2.0.1, deer: BioProject PRJNA324173, assembly CerEla1.0, goat: BioProject PRJNA340281, assembly ARS1, sheep: BioProject PRJNA179263, assembly Oar_v4.0) using STAR (Version 2.5.3a). Quality-trimmed and unmapped reads were assembled via SPAdes (Version 3.11.1).

Resulting scaffolds were then aligned to virus databases (Genbank and RefSeq viral nucleotide sequences downloaded on 12th of April 2018, UniProt viral amino acid sequences downloaded on 12th of March 2018) using BLASTn (Version 2.7.1+, default settings) and DIAMOND (Version 0.9.10, default settings). In order to exclude false positives, the scaffolds with a virus hit were aligned to an in-house non-viral database consisting of archaeal, bacterial, fungal, mammal, and protozoal sequences. Scaffolds were considered false positive if they had a longer hit on a sequence of the in-house database compared to the virus databases or if they had a nucleotide hit of more than 10% of their own length to any sequence of the non-viral database. Scaffolds with hits to AstV on nucleotide and/or amino acid level were considered valid AstV hits. For further analysis, scaffolds with a length over 5,550 nt and the three AstV-typical ORF including the ribosomal frameshift sequence between ORF1a and ORF1b were selected.

Rapid amplification of cDNA ends

In order to complete the inchoate 5′ ends of the five almost full-length scaffolds, a rapid amplification of cDNA ends (RACE) was performed. This 5′ RACE was carried out with a RACE-System (Invitrogen™, Carlsbad, CA, USA) according to the manufacturer’s instructions except of the usage of SuperScript™ III Reverse Transcriptase (Invitrogen™, Carlsbad, CA, USA) instead of SuperScript™ II Reverse Transcriptase (Invitrogen™, Carlsbad, CA, USA). Nested PCR was performed with gene specific primers (Table S1) and Taq DNA Polymerase with Standard Taq Buffer (New England BioLabs, Inc., Ipswich, MA, USA) as described by the manufacturer and an annealing temperature of 55 °C.

After visualization on a 1% agarose gel, PCR-products were excised of the gel and purified using NucleoSpin Gel and PCR Clean-up system (Macherey-Nagel, Düren, Germany) according to the manufacturer’s protocol.

Each 3 μL of the purified PCR products was Sanger-sequenced using the BigDye® Terminator v3.1 Cycle Sequencing Kit (Applied Biosystems, Foster City, CA, USA) in a 3730 DNA Analyzer (Applied Biosystems, Foster City, CA, USA) according to the manufacturer’s protocol. Sequence data were analyzed using the Geneious software package (Biomatters, Auckland, New Zealand, Version 10.2.6), trimmed with an error probability limit of 0.05 on both ends and aligned to the respective scaffolds.

Phylogenetic analysis

Capsid precursor sequences of 36 AstV strains were used for phylogenetic analysis and included AstV strains of various ruminants and our newly discovered sequences. Based on these sequences, a phylogenetic tree was constructed using the maximum likelihood method with 1,000 bootstrap replicates based on the Le_Gascuel_2008 with Freqs. model (Le & Gascuel, 2008) in MEGA 7 (Version 7.0.26). The model was chosen using the Find Best DNA/Protein Models option in MEGA 7.

Recombination analysis

Putative recombination events were assessed using the Recombination Detection Program (RDP4, Version 4.94) (Martin et al., 2015), following the RDP4 Instruction Manual (http://web.cbio.uct.ac.za/∼darren/RDP4Manual.pdf) and with the highest acceptable p-value set to 0.01. Recombination events were considered only when involving at least one of our newly generated sequences and having a highest acceptable p-value of 0.01 with all of the following methods: RDP (Martin & Rybicki, 2000), GENECONV (Padidam, Sawyer & Fauquet, 1999), Bootscan (Martin et al., 2005), Maxchi (Smith, 1992), Chimaera (Posada & Crandall, 2001), SiScan (Gibbs, Armstrong & Gibbs, 2000) and 3Seq (Lam, Ratmann & Boni, 2018).

Confirmation of newly discovered astrovirus sequences

To confirm novel AstV sequences, discovered by NGS, a RT-qPCR using specific primer-probe-combinations for each putative virus-candidate was applied. Specific primers and probes were designed for each of the almost full-length scaffolds using the Geneious software package (Biomatters, Auckland, New Zealand, Version 11.1.4). The primer and probe sequences with the respective nucleotide positions of the target are provided in Table S1.

All RT-qPCR reactions were performed using the TagMan™ Fast Virus 1-Step Master Mix (Applied Biosystems, Foster City, CA, USA) in 10 μL reactions according to the manufacturer’s instructions with a final concentration of 500 nM for each primer and 125 nM for each probe. For each reaction 2 μL RNA were added to 8 μL Master-Mix. The RT-qPCR was performed using a CFX96™ Real Time System on C1000 Touch™ Thermal Cycler (BioRad, Hercules, CA, USA), with the following cycle settings: 10 min, 45 °C; 10 min, 95 °C, and 40 cycles (15 s, 95 °C; 20 s, 61 °C; 30 s, 60 °C). After each elongation step, fluorescence was measured and analyzed with the CFX Maestro software (Version 4.1.2433.1219, BioRad, Hercules, CA, USA) with an auto calculated baseline threshold. Samples with a ct-value <35 were defined as positive.

Retrospective screening of samples for newly discovered astrovirus-candidates

After confirmation of all newly discovered AstV-candidates, all remaining 159 samples were re-tested using the RT-qPCR protocols as described above. For each run, a non-template control was used as negative control. The respective confirmed samples served as target-specific positive controls.

Results

Detection of neurotropic astroviruses BoAstV-CH13 and OvAstV-CH16 by RT-qPCR

Quantitative RT-PCRs were performed for the detection of neurotropic AstV. Positive controls showed ct-values on average of 22.73 (22.62–22.86) for BoAstV-CH13 and 23.22 (21.74–24.61) for OvAstV-CH16, respectively. All negative controls were scored negative. All 164 fecal samples had ct-values >35 or were reported as “undetectable” and were therefore diagnosed as negative for BoAstV-CH13 and OvAstV-CH16.

Detection of various astroviruses using Pan-astrovirus RT-PCR

To ensure the effective implementation and to establish positive controls, the protocol was first applied to brain-extracts of BoAstV-CH13 and OvAstV-CH16 positive animals. These samples showed a strong and clear band with the expected size of 450 bp. Nineteen of the 164 tested fecal samples derived from sheep (n = 6: S1–S6), goats (n = 8: G1–G8), alpaca (n = 3: A1, A3, A4) and deer (n = 2: D2, D5) showed a band at 450 bp and were therefore defined as Pan-AstV RT-PCR-positive.

All 19 samples, which were positive tested in Pan-AstV RT-PCR were further processed for NGS. Therefore, RNA was de novo extracted, capsid-bound viral genomes were relatively enriched and free nucleic acids were depleted using benzonase treatment.

NGS and bioinformatics analysis

Sequencing in paired-end mode (2 × 150bp) generated 45′455′879–233′701′511 reads per sample and raw reads were deposited in the European Nucleotide Archive (Accessions ERR3143214–ERR3143223). In 12 samples, a total of 29 scaffolds ranging from 521 to 6255 bp in size, showed AstV hits in the bioinformatics analysis on amino acid level, with sequence identity to known AstV sequences ranging from 39.3% to 96.4% (Table S2). The vast majority of these hits was to AstV strains identified in bovine fecal samples. Strikingly, in one sample of a sheep (sample S3), a 967 nt scaffold had a hit on the neurotropic BoAstV-CH13, isolate 42,799 (Bouzalas et al., 2016) on nucleotide (identity 83.4%, hit length 296 nucleotides) as well as on amino acid (identity 57.2%, hit length 306 amino acids) level. The k-mer coverage of this scaffold was <×4, which indicates relatively low RNA concentrations. The sequence mapped to the 3′ half of ORF2, which encodes the hypervariable part of the capsid precursor protein (Babkin et al., 2012).

For the remaining AstV hits, we further analyzed scaffolds with a minimum length of 5,500 nt and covering parts of all three AstV-typical ORFs. One such scaffold was identified in two sheep and in three goats (Table S2). These tentative AstV scaffolds were designated CapAstV-G2.1, -G3.1 and -G5.1 and OvAstV-S5.1 and -S6.1. Each of these scaffolds showed the ribosomal slippery sequence (5′-AAAAAAC-3′) at the ORF1a/1b junction and an overlap between ORF1b and ORF2. The translated nsp1ab sequences had an identity between 71.7% and 75.4% to the next best hit while the capsid protein precursor ranged from 57.5% to 73.7% (Table 1). Best hits were AstV strains previously described in cattle (Nagai et al., 2015; Tse et al., 2011), deer (Smits et al., 2010) and takin (Guan et al., 2018). Taken together, these findings clearly support the notion that these five scaffolds represent complete or almost complete viral genomes of novel MAstV strains.

Table 1 Discovery of five novel full-length astrovirus genomes in sheep and goats.

	Full genome best hit (fullgenome-accession)	Full genome identity (%)	ORF 1ab best hit non-structural protein (1ab-BLASTP-accession)	ORF1ab identity (%) non-stuctural protein	ORF2 best hit capsid protein (2-BLASTP-accession)	ORF2 Identity (%) capsid protein	
CapAstV-G2.1 (MK404645.1)	Bovine astrovirus JPN/Hokkaido11-55/2009 (LC047790.1)	76.6	BoAstV/JPN/Hokkaido11-55/2009 (BAS29607.1)	75.4	BoAstV/JPN/Hokkaido11-55/2009 (BAS29609.1)	73.7	
CapAstV-G3.1 (MK404646.1)	Bovine astrovirus B170/HK (HQ916314.1)	69	Sichuan takin astrovirus (YP_009480536.1)	71.7	Bovine astrovirus B170/HK (YP_009010954.1)	63.9	
CapAstV-G5.1 (MK404647.1)	Sichuan takin astrovirus (NC_037655.1)	70.9	Bovine astrovirus B76-2/HK (YP_009010946.1)	75.1	Astrovirus deer/CcAstV-1/DNK/2010 (ADO67579.1)	57.7	
OvAstV-S5.1 (MK404648.1)	Bovine astrovirus B76-2/HK (HQ916317.1)	70.5	Bovine astrovirus B76-2/HK (YP_009010946.1)	73.8	Astrovirus deer/CcAstV-1/DNK/2010 (ADO67579.1)	59.0	
OvAstV-S6.1 (MK404649.1)	Sichuan takin astrovirus (NC_037655.1)	70.8	BoAstV/JPN/Ishikawa24-6/2013 (BAS29598.1)	74.6	Astrovirus deer/CcAstV-1/DNK/2010 (ADO67579.1)	57.5	
Note:

Best hits of the bioinformatics pipeline on nucleotide (full-genome) and amino acid (ORF 1ab, ORF2) level are presented separately. Genbank accession number are provided in brackets. BoAstV, bovine astrovirus; CapAstV, caprine astrovirus; OvAstV, ovine astrovirus.

Rapid amplification of cDNA ends

In four of the five samples analyzed, the 5′ RACE did work and Sanger sequencing added to the 5′ end of the respective almost full AstV scaffolds (G3.1, G5.1, S5.1, S6.1). This resulted in the completion of these four scaffolds, which, together with CapAstV-G2.1, were deposited on GenBank (Accessions: CapAstV-G2.1 (MK404645.1), CapAstV-G3.1 (MK404646.1), CapAstV-G5.1 (MK404647.1), OvAstV-S5.1 (MK404648.1), and OvAstV-S6.1 (MK404649.1)).

Phylogenetic analysis

To assess the genetic relationship of the newly discovered strains and partial sequences to other known MAstV, a phylogenetic comparison of all generated putative genomes to representative MAstV genotype species and so far unclassified strains with high sequence similarity based on the capsid precursor protein sequences (Fig. 2) was conducted. For the goats, all three novel CapAstV strains clustered in different branches of the phylogenetic tree. CapAstV-G2.1 and CapAstV-G3.1 are both related to two different BoAstV strains: BoAstV/JPN/Hokkaido 11-55/2009 (p-dist. 0.293) and BoAstV-B170/HK (p-dist. 0.361), respectively. CapAstV-G5.1 is very similar to the new OvAstV strains S5.1 (p-dist. 0.236) and S6.1 (p-dist. 0.027), clustering together in one clade of the phylogenetic tree. They cluster with a p-distance of 0.419 to the closest related AstV strain CcAstV-1/DNK/2010, which has been detected in feces from deer (Smits et al., 2010) (Table 1; Fig. 2). Partial sequences generated from goat feces cluster together with the here described CapAstV/OvAstV or with enterotropic strains from other ruminants. In the partial sequences generated from sheep and deer feces, the situation is similar to the one described in goats. As previously indicated, one partial sequence, S3.1, is not clustering close to other enterotropic AstV but rather near the neurotropic BoAstV-CH13/NeuroS1 cluster. In samples with multiple AstV hits, partial sequences cluster at different positions of the phylogenetic tree. OvAstV-S5.1 clusters together with CapAstV-G5.1/OvAstV-S6.1 and S4.1, while scaffold S5.2, detected in the same sample, clusters together with sequences from sample G8 (G8.3/G8.5).

Figure 2 Phylogenetic analysis of novel small ruminant astroviruses.

Phylogenetic analysis using the maximum likelihood method, based on 76 amino acid sequences of the capsid precursor protein of selected astrovirus strains together with the ones generated by the bioinformatics pipeline, with the sheep (blue), goat (green), and deer sequences (purple) from this study marked. GenBank accession numbers are provided in brackets. Filled rhombi indicate encephalitis-associated strains described in animals. Capsid protein precursor sequences translated from scaffolds with less than three identified ORFs are marked with an asterisk. AvAstV, avian astrovirus; BoAstV, bovine astrovirus; CcAstV, deer astrovirus; DromAstV, dromedary astrovirus; HuAstV, human astrovirus; MiAstV, mink astrovirus; PoAstV, porcine astrovirus; StAstV, sichuan takin astrovirus; WBufAstV, water buffalo astrovirus; YakAstV, yak astrovirus.

Recombination analysis

Knowing about the high tendency toward recombination in RNA viruses (Matsui et al., 1998) and previous reported indications of interspecies transmission of AstV based on recombination analysis (De Battisti et al., 2012; Lan et al., 2011), it was decided to determine whether such recombination events may have occured in the newly discovered strains. Subsequent to the phylogenetic analysis, the five new AstV sequences were analyzed together with selected AstV strains for plausible recombination events. Three putative recombination events were reported (Table S4). Between the three strains OvAstV-S6.1, CapAstV-G5.1 and OvAstV-S5.1, two recombination events were reported, one starting at position 4,062 and ending at position 6,114 (recombination event 1) and the other starting at position 1,737 and ending at position 2,443 (recombination event 3) (Fig. 3A) in OvAstV-S6.1. All three sequences may be the recombinant. Recombination event 3 was predicted to the overlap between ORF1a and ORF1b, whereas recombination event 1 was predicted for almost the entire ORF2. Another putative recombination event (2) was identified between BoAstV-GX27/CHN/2014 and BoAstV/JPN/Hokkaido11-55/2009 or closely related sequences as potential parental sequences, with the 5′ breakpoint at position 36 and the 3′ breakpoint at position 1,154 resulting in CapAstV-G2.1 as the recombinant (Fig. 3B). Identities on nucleotide and amino acid level of the strains involved in the recombination events can be found in Table S3.

Figure 3 Recombination analysis of newly discovered full-length astrovirus genomes.

(A) Predicted recombination events 1 and 3 between CapAstV-G5.1, OvAstV-S5.1 and OvAstV-S6.1, where all three sequences may be the resulting recombinant. (B) Predicted recombination event 2 with CapAstV-G2.1 as the recombinant, BoAstV/GX27 as the minor parent and BoAstV/JPN/HK as the major parent. Plots were constructed using the RDP Method graphical output in RDP4. Nucleotide positions within the astrovirus genome are depicted on the axis of abscissas in kb. Red bars schematically indicate the parts of the genomes involved in the recombination events. For comparison, the astrovirus genome organization is presented at the bottom. BoAstV/JPN/HK, Bovine astrovirus genomic RNA, nearly complete genome, strain: BoAstV/JPN/Hokkaido11-55/2009 (LC047790.1) BoAstV/GX27, Bovine astrovirus strain BAstV-GX27/CHN/2014, complete genome (KJ620980.1) CapAstV, Caprine Astrovirus OvAstV, Ovine Astrovirus.

Confirmation of newly discovered astrovirus sequences

Using specific RT-qPCRs, all five newly discovered AstV-candidates could be confirmed with the following ct-values: G2, 27.65; G3, 25.26; G5, 26.99; S5, 25.77; S6, 25.28. The non-template control remained negative for all targets. Sample G5 showed a positive signal (ct-value 27.09) for the primer-probe-pair S6.1, which is not surprising, considering the fact that these two samples share highly similar sequences. In all other samples, no cross-reactivity could be detected between the individual targets.

Retrospective screening of samples for newly discovered astrovirus-candidates

All 159 samples remained negative for the primer-probe combinations G3.1, G5.1, S5.1, and S6.1. Surprisingly one sample originating from a goat, scored positive for G2.1 (ct-value 29). Based on information from the original submission site (Parasitology, Vetsuisse Bern), it could be found that this sample originated from the same holding as the sample G2.

Discussion

This study presents the discovery and molecular characterization of putatively new AstV genotype species in sheep as well as in goats. These novel AstV show a broad genetic diversity compared to AstV affecting other species. The present results support interspecies transmission of AstV between goats and sheep, as well as recombination events between AstV affecting sheep, goats and cattle.

The aims of the study were to investigate the potential shedding of neurotropic AstV in small ruminants’ feces, to discover and to analyze unknown AstV, and examine a putative interspecies transmission. Therefore, all available samples first were screened for the neurotropic AstV BoAstV-CH13 as well as BoAstV-CH15/OvAstV-CH16. Three different methods were then applied, first, highly specific RT-qPCR, second, a wide-spectrum RT-PCR, and third, NGS of all samples interpreted as positive after RT-PCR. None of these samples could be defined as positive for BoAstV-CH13 or BoAstV-CH15/OvAstV-CH16. Thus, there is no evidence that neurotropic AstV are part of the intestinal virome of small ruminants. Noteworthy, limitations of this study are the relative small number of samples investigated, the unknown disease status of the animals and the focus mainly on sheep and goats due to a lack of available samples from other ruminant species. A related study targeting feces and brain of cattle was reported previously (Oem & An, 2014).

Screening for various and potentially unknown AstV was done with a heminested RT-PCR that targets a conserved sequence at the 3′ end of ORF1b. However, RNA viruses, and in particular AstV, are predicted to undergo 3.4 × 10−3 mutations per nucleotide per genomic replication (Duffy, Shackelton & Holmes, 2008) and are genetically relatively diverse. Therefore, this RT-PCR uses highly degenerated primers, which has the limitations that (i) it may lack sensitivity; due to absence or inefficiency of primer binding to divergent AstV sequences and that (ii) unspecific primer binding may result in amplification of non-AstV sequences. On the one hand, as only RT-PCR positive samples were submitted to NGS, fecal samples containing divergent AstV strains may have been classified as false negative and therefore might have been excluded from further analysis. On the other hand, false positive samples may have been further analyzed by NGS. Indeed, seven RT-PCR positive samples did not reveal any AstV hit in the bioinformatics pipeline. This discrepancy may also be related to the sample pretreatment procedure prior to NGS-submission.

Beside the discovery and investigation of four full-length and one almost full-length AstV sequence, NGS and bioinformatics analysis resulted in the detection of 24 additional scaffolds, which covered only parts of the astroviral genome (Table S2).

One scaffold that was identified in a sheep sample (S3.1) had its best hit on the ORF2 region of BoAstV-CH13 (Table S2) and clustered near the BoAstV-CH13/NeuroS1 cluster in the phylogenetic tree. Attempts to confirm the sequence of this scaffold by RT-PCR, cloning and further sequencing approaches remained unsuccessful. This may be due to the low viral load or PCR inhibitors in small ruminants’ feces that interfere with amplification. The importance of this finding is so far unclear. With an identity of 57% on the amino acid level to BoAstV-CH13, this putative AstV would be part of the so-called human-mink-ovine-like clade, which also comprises the fecal OvAstV-1 strain as well as the vast majority of encephalitis-associated strains in humans and animals (Kapoor et al., 2009; Reuter, Pankovics & Boros, 2018). Given the fact that only a part of the sequence (967 nt) could be determined, it is still possible that this virus shows varying identities depending on the genome section compared to other viral sequences. However, the importance of a potential occurrence of a novel and divergent neurotropic AstV in this fecal sample could not be definitively elucidated.

For the remaining 23 scaffolds, best hits were consistently to bovine strains and, unexpectedly, not to previously described ovine (OvAstV-1 and OvAstV-2) (Reuter et al., 2012; Snodgrass & Gray, 1977) and deer AstV strains (Smits et al., 2010), respectively. Indeed, sequence similarities compared to best hits in the database entries on protein-level were between 39.3% and 96.4 % (median 72.7%). A total of 15 scaffolds covered at least a part of the ORF2 and therefore underwent phylogenetic analysis based on the capsid protein precursor sequence together with the five newly identified full-length or almost full-length AstV genomes (Fig. 2). This analysis clearly points out the broad diversity of putative AstV detected in this study. Novel sequences were not only discovered in sheep and goat, but also in deer (D2.1, D5.1), interestingly showing a low similarity to known AstV in deer (CcAstV-1/DNK/2010, CcAstV-2/DNK/2010). In addition, several sequences found in sheep show a clearly higher identity to bovine AstV than to ovine AstV (e.g., S6.3* to BoAstV/JPN/Hokkaido12-25/2009). Taken together, these results clearly suggest a much larger diversity of AstV in small ruminants and deer than known to date. Moreover, the occurrence of multiple phylogenetically different AstV-like scaffolds in the same sample in most of these animals (e.g., in G1, G8 and S6) supports that AstV coinfections are frequent.

Five novel, including four full-length and one almost full-length AstV sequences were discovered: two in sheep and three in goats. Noteworthy, this study reports for the first time an AstV infection in goats. Based on the genetic distances of the capsid protein precursor sequences to the closest related strains, these caprine and OvAstV are grouped into three different genotype species. According to the ICTV 9th report 2011 (Bosch et al., 2011), amino acid differences between AstV genotype species are >0.338. Based on this criterion, the strains described in this study belong to three AstV genotype species: (i) CapAstV-G2.1 together with BoAstV/JPN/Hokkaido 11-55/2009, (ii) CapAstV-G3.1 on its own, and (iii) CapAstV-G5.1 together with OvAstV-S5.1 as well as OvAstV-S6.1. While CapAstV-G2.1 belongs to the same genotype species as BoAstV/JPN/Hokkaido 11-55/2009, CapAstV-G3.1 is sufficiently diverse to build a new genotype species on its own. In particular, CapAstV-G5.1 and OvAstV-S6.1 were very similar to each other, not only in the capsid protein sequence (97.3%), but also in the predicted non-structural proteins (90.3%) and on full-length genome nucleotide level (87.6%) (Table S3). The high similarity between these viruses raises the question of transmission events between sheep and goats and is reminiscent of the situation observed for BoAstV-CH15 and OvAstV-CH16 in AstV-associated encephalitis (Boujon et al., 2017).

A retrospective screening of the whole sample panel targeting the five newly discovered AstV candidates resulted in only one additional positive sample for target G2.1. This sample originated from the same holding as animal G2 and was collected at the same day. While this screening indicated an overall low prevalence of the newly discovered AstV candidates, it appears plausible that these viruses transmit between animals and that the in herd prevalence may be much higher.

Because recombination between viruses requires infection of the same host cell, the existence of interspecies transmission events are further supported by predicted recombination events (Wolfaardt et al., 2011). In the present study, recombination events were forecast between ovine and caprine strains as well as between caprine and bovine strains. Not only was recombination predicted for OvAstV-S6.1, OvAstV-S5.1 and CapAstV-G5.1, but also for CapAstV-G2.1 and two divergent bovine AstV (BoAstV/JPN/Hokkaido11-55/2009, BAstV-GX27/CHN/2014), suggesting that these viruses (or close relatives to those strains) shared the same host at some time point during evolution (Fig. 3; Table S4). Similar recombination events have been proposed between a porcine AstV and HAstV-3 (Ulloa & Gutierrez, 2010). In addition, the transmission of fecal AstV between cattle and roe deer has been suggested (Smits et al., 2010; Tse et al., 2011). Taken together, all these data challenge the assumption that mammalian AstV are strictly host specific. The host-specificity of Astroviridae was already challenged in previous studies (Chu et al., 2010; Karlsson et al., 2015; Rivera et al., 2010).

Conclusion

The question whether enterotropic AstV can cause disease in small ruminants remains so far unresolved. Due to the mainly unknown health status of the tested animals, the importance of AstV occurring in small ruminants’ feces remains so far unclear and needs to be further investigated. Still, this study describes five novel AstV discovered in small ruminants, including the first description of an AstV in goats and gives new insights into the frequency and diversity of AstV in ruminant species.

Supplemental Information

Supplemental Information 1 Primer and probe sequences.

Click here for additional data file.

Supplemental Information 2 Best hits of sequences generated via NGS and bioinformatics analysis.

Click here for additional data file.

Supplemental Information 3 Pairwise distances of strains involved in recombination events.

Click here for additional data file.

Supplemental Information 4 Overview of breakpoint positions detected by recombination analysis.

Click here for additional data file.

Supplemental Information 5 Sequence data, Genbank submission.

Click here for additional data file.

The authors thank the Institute of Parasitology Vetsuisse Faculty, University of Bern (Bern, Switzerland), especially W. Basso, for providing the fecal samples and S. Schenk as well as M. Fragnière of the Next Generation Sequencing Platform of the University of Bern for performing the high-throughput sequencing experiments.

Additional Information and Declarations

Competing Interests

Author Contributions

DNA Deposition

Data Availability

New Species Registration

The authors declare that they have no competing interests.

Ronja V. Kauer conceived and designed the experiments, performed the experiments, analyzed the data, contributed reagents/materials/analysis tools, prepared figures and/or tables, authored or reviewed drafts of the paper, approved the final draft.

Michel C. Koch performed the experiments, analyzed the data, contributed reagents/materials/analysis tools, prepared figures and/or tables, authored or reviewed drafts of the paper, approved the final draft.

Melanie M. Hierweger performed the experiments, contributed reagents/materials/analysis tools, authored or reviewed drafts of the paper, approved the final draft.

Simea Werder performed the experiments, contributed reagents/materials/analysis tools, approved the final draft.

Céline L. Boujon performed the experiments, contributed reagents/materials/analysis tools, authored or reviewed drafts of the paper, approved the final draft.

Torsten Seuberlich conceived and designed the experiments, contributed reagents/materials/analysis tools, prepared figures and/or tables, authored or reviewed drafts of the paper, approved the final draft.

The following information was supplied regarding the deposition of DNA sequences:

Data are available at NCBI Genbank under accession numbers MK404645.1, MK404646.1, MK404647.1, MK404648.1, and MK404649.1. The newly discovered, almost full length astrovirus scaffolds, were named CapAstV-G2.1 (GenBank accession number: MK404645.1), -G3.1 (MK404646.1), -G5.1 (MK404647.1), OvAstV-S5.1 (MK404648.1), and -S6.1 (MK404649.1).

The following information was supplied regarding data availability:

Data is available at the European Nucleotide Archive under accession numbers ERR3143214–ERR314232.

https://www.ebi.ac.uk/ena/data/view/PRJEB31037

The following information was supplied regarding the registration of a newly described species:

Family name: Astroviridae

Genus name: Mamastrovirus

Species names: Caprine Astrovirus G5.1 and Caprine Astrovirus G3.1

The proposed new species Caprine Astrovirus G5.1 and Caprine Astrovirus G3.1 have been submitted to the ICTV for consideration. These names are only valid and official after the ICTV has approved the names, and they have been ratified by the membership: (URL pending)

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
