# Peer review of "Discovery of novel astrovirus genotype species in small ruminants"

_PeerJ, doi:10.7717/peerj.7338_

## Round 0.1 · original submission · Minor Revisions

Please regard that some in-depth explanation are needed to clarify portions of the materials and methods as well as for the results.

Reviewer 1 ·

Basic reporting

Line 41 as well as in the figure legend of fig. 3: “five newly discovered full-length genome sequences” indicates that the authors determined the complete genomic sequences of the five sheep and goat astroviruses, although there are otherwise contradictory information found in the manuscript about completeness or even the exact lengths of sequences of the identified novel sheep and goat astroviruses. e.g. lines 252 – 253: “these five scaffolds represent complete or almost complete viral genomes of novel MAstV strains” or in lines 346-347: “Five novel, almost full-length astrovirus sequences were discovered…”

Lines 82-84: Additional animal species other than humans and ruminants from which neurotropic astroviruses were also detected (e.g. minks and swine) should be mentioned at least in the introduction because the neurotropic astroviruses identified from those animals are highlighted in figure 2.

Lines 84-87: The source materials (e.g. brain or CNS samples etc…) of these neurotropic astroviruses should be included here.

Lines 223-226: Please consider that this section should be moved to the Detection of various astroviruses using Pan-astrovirus RT-PCR (line 208) section.

Lines 267 – 271: Please consider that this section should be moved to the Discussion section

Figure 2: Please consider including “clades” to the phylogenetic tree according to ICTV taxonomy guidelines (https://talk.ictvonline.org/ictv-reports/ictv_9th_report/positive-sense-rna-viruses-2011/w/posrna_viruses/248/astroviridae-figures) Please consider to use different colors of existing and newly discovered ovine strains (OvAstV). Same color of previously known and novel ovine astrovirus strains is confusing.

genbank submissions: Why tissue_type="brain" could be found in all submission files? According to the manuscript these sequences were identified from fecal samples.

Experimental design

Lines 126-128; lines 132-133 and supplementary table 1: The positions of the primers and probes should be described in more detail, e.g. nt positions on a reference genome or designations of the targeting genomic regions (e.g. RdRp, protease, capsid-conserved/hypervariable part…) should be specified.

Lines 226 – 230: Please consider that the results most likely originated from contamination should be omitted because it is not relevant to the original scope of the manuscript

Lines 285 – 286 as well as in the figure legend of fig. 3: Please consider to include the nt/aa identities between the CapAstV-G2.1 and BoAstV/GX27 as well as between CapAstV-G2.1 and BoAstV/JPN/HK. It is most likely that the exact parental sequences are not BoAstV/GX27 and BoAstV/JPN/HK but other astroviruses related to these viruses. Therefore this section should be rephrased like “Another putative recombination event (2) was identified between BoAstV-GX27/CHN/2014 and BoAstV/JPN/Hokkaido11-55/2009 or closely related sequences as potential parental sequences”

Validity of the findings

Line 290 – 291: Although some of the identified ovine and goat astroviruses are fulfilled the criterion of novel species but these viruses are still candidates therefore this sentence should be rephrased accordingly.

Reviewer 2 ·

Basic reporting

In this study, Kauer and colleagues screened a panel of 164 fecal samples from small ruminants in Switzerland for known encephalitis-associated virus astrovirus strains in sheep and cattle. Furthermore, they tested the samples by a published heminested panastrovirus RT-PCR for other astroviruses, sequenced positive samples and performed recombination analysis.
The manuscript is well written and the conclusions stated by the authors are comprehensible. As different astroviruses in encephalitic animals were described independently several times in the last years and nothing is known about the excretion or pathogenesis, the topic of this manuscript is relevant for a better understanding of these viruses.

Experimental design

The original primary research presented in this work is within the scope of the journal. The research question is well defined, relevant & meaningful.
Nonetheless, the single sequences obtained by NGS should be verified by RT-(q)PCRs and a test for the frequency of the detected new genotypes would strengthen the work.

Validity of the findings

The authors could not detect any so far known neurotropic astroviruses in the sample panel but were able to discover new astrovirus genotypes in small ruminants which further broaden the genetic diversity of this virus family.
Unfortunately, the number of investigated samples is low and no information about the health status of the animals is available.
The analysis of recombination events indicates possible interspecies transmission, highlights the possible zoonotic potential and thus once again shows that the former opinion of species-specific astroviruses appears to be wrong.

Additional comments

Line 17: Space character missing.
Line 43: Explain why unexpected? Divergent Astroviruses in feces of animals have been shown before.
Line 48: Space character missing.
Line 135: Were internal controls used to check extraction efficacy and quality of sample materials?
Line 157: Revise word order.
Line 183: What percentage of the scaffold consists of sequence information? Where are the gaps located? A figure in the supplements representing the genome coverage would be helpful (at least for the five large scaffolds).
Line 224: Could it be that the benzonase treatment affects not only the amount of free nucleic acids but also reduces to some extent the overall concentration of astroviral RNA within the capsids? What sequences were generated from the RT-PCR amplicon of the other ten samples before benzonase treatment which scored negative after the treatment?
Line 299: RT-qPCR. Please check the spelling throughout the manuscript.
Line 309: Were the other sequences found by NGS confirmed with specific RT-(q)PCRs? This should be done.
Line 346: Further information about the frequency of the five newly detected astrovirus genotypes would improve the manuscript. Would it be possible to screen the panel of feces samples with RT-PCR systems for theses variants? The panastrovirus heminested PCR is maybe not that sensitive and you may have missed positive samples.
Figure 2: Please include all encephalitis-associated strains described in animals.

·

Basic reporting

No comment.

Experimental design

I think further effort could have been made to extend the sequence information of some of the viruses and that partially sequenced viruses could have also been included in the phylogenetic analysis (see below for details).

Validity of the findings

Recombination analyses need some clarification (see below for details).

Additional comments

Kauer et al. report the identification of several potentially novel astroviruses of ruminants, although they only obtained extended sequence information for 5 of them. These novel viruses belonged to 3 species and affected goat and sheep. There was evidence for cross-species transmission and viral recombination. The paper is well written and properly structured, the objective is well-defined and proper methods were used, analyses are thorough, and results properly discussed. However, there are some points that, in my opinion, deserve a more in-depth analysis/explanation.

1. First of all, obtaining as much sequence information as possible is crucial for this kind of studies. The authors have chosen to consider for an in-depth investigation only viruses for which the complete coding sequence was determined. However, I believe that more effort could have been made to try and extends obtained scaffold sequences (unless it was attempted but not described). For example, extended sequences could be obtained by designing primes and connecting fragments, and through genome-walking or RACE methods. I realize that such laboratory investigations are not always possible as they are laborious and costly, but bioinformatics methods can be used for this purpose too. For example, although blast and diamond settings were not provided, did the author try to use a less strict search to identify viral fragments with a lower identity to reference strains? Also lose-setting mapping and cross-assembly (between different samples) could help in identifying viral fragments and their relationships (obviously such assemblies would have to be PCR-confirmed). For example, from a quick analysis, it looks like samples G8 and S5 contain the same virus and I could successfully assemble fragments G8-3/4/5 and S5-2 into a ~2.7Kb contig. This study would benefit a lot from obtaining further complete genomes.

2. Although for some viruses only partial sequence was obtained, for some of them a sizable portion of the genome was obtained that allows to perform meaningful phylogenetic analysis. I think partially sequenced strains should not be dropped and, although definitive classification cannot be provided, an additional phylogenetic analysis should be performed to describe partially sequenced strains. These strains, may not have the requirements to be classified as novel species, but they are still interesting (I blasted some of them and many may be additional new viruses) and their phylogenetic relationship to other astroviruses should be investigated. Such analysis would also help to prove the point you make at line 335 about co-infections: you do have overlap between different fragments from the same sample but you never show how similar they are to each other (for example, you clearly have three different viruses in sample G1).

3. Recombination analysis. Aren’t the setting used a bit strict? Normally you would choose events detected by at least 2-3 methods, not by all. Can you justify your choice? Furthermore, multiple contigs were found in samples involved in recombination. Did you make sure that only one virus was present in these samples and that recombination patterns were not due to artificial assembly of reads? Were recombination events confirmed by building trees with portions of the alignments included between breakpoints? If not, this should be done.

Minor.
- Line 26. It should be “aims” and “were”.
- The introduction is very well written and complete. However, I think that a clearer classification of bovine/ovine astroviruses at a species level could be provided to help the reader distinguishing between strains (e.g. which strains are included in species MAstV-13 and which other likely belong to different species). More information can be found in https://doi.org/10.3390/v9050102.
- Figure 1. There are some weird arrows in the middle panel coming down from the first box.
- Supplementary Table 1. I would add references to papers including primers previously published.
- Lines 150 and 153. Are you sure primers were used at a final concentration of 10uM? Normally protocols use a final primer concentration of 0.05-1uM. Please, verify.
- Line 157. This sentence is a bit weird and English should be checked.
- Lines 176 and 216. I think the correct form is BLASTn (n in lower case).
- Line 163. I suggest using a more elegant title without abbreviations.
- Line 164. To facilitate reading, I think a comma after “(NGS)” should be introduced.
- Line 189. How was this model chosen, based on a modeltest analysis or on literature?
- The first three paragraphs of the results can be combined into one unique paragraph which summarizes the information contained in these three paragraphs, which also contain irrelevant information (e.g. the first paragraph could be summarized into “None of the samples were positive for BoAstV-CH13 and OvAstV-CH16 ”. Also, lines 209-211 can be removed).
- Lines 215-220. First of all, how did you get a >600nt sequence when the amplicon was 450nt? Were some bands bigger or were you sequencing something else? Can it be that this number was reported wrong (according to supplementary table 2, no sequence was longer than ~300nt)? Secondly, blasting a 6nt seems a bit absurd! These results clearly indicate that, in many cases, there were some issues with sequencing and the question “why these amplicons were not cloned before sequencing” comes into mind. I strongly suggest removing this analysis and simply stating that 19 samples were astrovirus-positive and those were used for NGS (results of this sequencing are ignored anyway as all samples were used for NGS). (The same is true for the results of the PCR performed after pre-treatment).
- Line 223. “de novo” should be italicized.
- Line 243. I assume your selection criteria was that sequences should cover the full-length of the three ORF. You should specify if this is the case because it is not entirely clear.
- Line 253. I would say these represent novel species as species demarcation criteria for astroviruses (if I am not wrong) is 75% aa identity in the capsid.
- Line 257. The word “isolate” is referred to viruses that have been isolated on cell-culture. I would use the word “strain” instead.
- Supplementary table 4. I would remove the last column as it is not informative. Also, the sentence below the table is a bit weird.
- Figure 2. I would mark more clearly the strains identified in this study and indicate clades on the tree (e.g. accepted species and/or HMO clade).
- Figure 3. Which of the methods used in RDP was used to generate these plots? This must be indicated.
- Line 335. You can only talk about co-infections if you compared the different contigs identified in the same sample and show they are different…
- Line378. There is a typo (astroviruses/in).
- Did I understand correctly that raw sequences were submitted to GeneBank? If so accession numbers should be provided in the text.

---

## Round 0.2 · accepted · Accept

Thank you, very much for this contribution. The final suggested edits can be addressed while in production

Reviewer 1 ·

Basic reporting

Lines 83 - 85 of the revised manuscript: As far as the reviewer knows the first a neurovirulent porcine astroviruse type 3 was discovered in 2017 by two independent research teams in the USA and Hungary (Arruda et al., 2017: Emerging infectious diseases, 23(12), 2097 and Boros et al., 2017: Emerging infectious diseases, 23(12), 1982). The two references given by the authors (Laurin et al. 2011; Reuter et al. 2012) described only enteric porcine astroviruses unrelated to CNS infections.

Experimental design

Adequate

Validity of the findings

All underlying data have been provided and the conclusions are well stated.

Additional comments

None. Nice work!

Reviewer 2 ·

Basic reporting

All necessary changes were done. No further comment.

Experimental design

All necessary changes were done. No further comment.

Validity of the findings

All necessary changes were done. No further comment.

Additional comments

All necessary changes were done.

·

Basic reporting

no comment

Experimental design

no comment

Validity of the findings

no comment

Additional comments

I am satisfied with the revisions. I found 2 typos and I have one remark:
- line 84: typo: astroviruse.
- line 424: typo. There is a wild "of" at the end of the line.
- lines 428-433. Given the fact that these viruses seem to be very species-unspecific and the fact that some are pretty divergent, I don't think you can't conclusively determine from which hosts parental viruses came from and also, likely, parental strains are not exactly those used for the RDP analyses, but relatives to those strains. I suggest rephrasing this paragraph.